# Tandem 13-Lipoxygenase Genes in a Cluster Confers Yellow-Green Leaf in Cucumber

**DOI:** 10.3390/ijms20123102

**Published:** 2019-06-25

**Authors:** Yin Ding, Wei Yang, Chenggang Su, Huihui Ma, Yu Pan, Xingguo Zhang, Jinhua Li

**Affiliations:** 1Key Laboratory of Horticulture Science for Southern Mountainous Regions, Ministry of Education, College of Horticulture and Landscape Architecture, Southwest University, No.2 Tiansheng Road, Beibei, Chongqing 400715, China; dd815988458@email.swu.edu.cn (Y.D.); shucaitomato@gmail.com (W.Y.); suchenggang@swu.edu.cn (C.S.); h2394978356@email.swu.edu.cn (H.M.); pany1020@swu.edu.cn (Y.P.); zhangdupian@swu.edu.cn (X.Z.); 2State Cultivation Base of Crop Stress Biology for Southern Mountainous land of Southwest University, Academy of Agricultural Sciences, Southwest University, Beibei, Chongqing 400715, China

**Keywords:** 13-lipoxygenase, gene cluster, yellow green leaf, cucumber, carotenoids

## Abstract

Some lipoxygenase (LOX) isoenzymes can co-oxidize carotenoids. Carotenoids are collectors of light energy for photosynthesis and can protect plants from reactive oxygen species and coloration. This study isolated the cucumber (*Cucumis sativus* L.) yellow-green leaf mutant (*ygl1*), which had yellow-green leaves with decreased chlorophyll synthesis, increased relative carotenoid content, and delayed chloroplast development. Genetic analysis demonstrated that the phenotype of *ygl1* was caused by a recessive mutation in a nuclear gene. The bulked segregants were resequenced, and the candidate *ygl1* locus identified was mapped to the 9.2 kb region of the chromosome 4. Sequence analysis revealed that *ygl1* encodes the tandem *13-LOX* genes in a cluster. Four missense mutations were found in four tandem *13-LOX* genes (*Csa4M286960*, *Csa4M287550*, *Csa4M288070*, and *Csa4M288080*) in the *ygl1* mutant, and the four *13-LOX* genes showed high similarity with one another. The transient RNA interference and virus-induced gene silencing of these genes simultaneously resulted in yellow-green leaves with a reduced amount of chloroplasts and increased relative carotenoid content, which were observed in the *ygl1* mutant. This evidence supported the non-synonymous SNPs (Single Nucleotide Polymorphism) in the four tandem *13-LOX* genes as being the causative mutation for the yellow-green leaves. Furthermore, this study provides a new allele for breeding cucumbers with yellow-green leaves and serves as an additional resource for studying carotenoid biosynthesis.

## 1. Introduction

Chlorophyll (Chl) content and chloroplast development are directly related to the efficiency of plant photosynthesis [1]. Elevated levels of carbon dioxide (CO_2_) result from leaf photosynthesis and increase crop yield potential [2]. The chloroplasts of green plants can fix CO_2_ and transform them into organic substances via photosynthesis. In chloroplasts, Chl molecules universally exist in photosynthetic organisms. Chl and carotenoids (Car) are the most abundant pigments that play important functions in plant chloroplast development and photosynthesis [3]. In addition to their well-established function as collectors of light energy for photosynthesis, the Car of higher plant chloroplasts might have an important structural role [3]. Chl molecules universally exist in photosynthetic organisms. They perform essential processes of harvesting light energy in the antenna systems and driving electron transfer in the reaction centers [4]. 

Chloroplast development and Chl and Car biosyntheses are correlated with organ coloration. Plant Car are red, orange, and yellow lipid-soluble pigments and are embedded in the membranes of chloroplasts. Their colors are masked by Chl in photosynthetic tissues [5]. Leaf color mutants are the ideal materials for exploring the mechanisms of Chl and Car biosynthesis and chloroplast development [1,6,7,8,9,10]. Plant leaf color variation is caused by genetic mutations that directly or indirectly affect the synthesis and degradation of Chl, thus changing the Chl content [11,12]. Studies on the molecular mechanisms of leaf color mutants are leading in model crops and field crops. Several genes associated with leaf color mutations have been cloned in *Arabidopsis* [13,14], rice [6,7,11,15], and tobacco [16], and these genes encode Chl synthesis. In breeding, leaf color variation can be used as marker traits to simplify seed breeding and hybrid production [17]; therefore, discovering and identifying plant Chl-deficient mutant genes have important theoretical significance and application value.

Lipoxygenases (LOXs), widely found in plants, fungi, and animals [18], are a large family of monomeric proteins with non-heme, non-sulfur, and iron cofactor containing dioxygenases that catalyze the oxidation of polyunsaturated fatty acids to yield hydroperoxides [19]. LOXs display diverse functions in mammals and plants. In mammals, they play important roles in the development of acute inflammation [20], and in cancer [21] and vascular biology [22], and they contribute to in vivo metabolism of endobiotics and xenobiotics [23]. Plant LOXs are primarily classified into two major classes, namely, 9- and 13-LOXs, on the basis of their positional specificity to oxygenate linoleic acids [24]. Furthermore, 13-LOXs can be classified into type I and type II on the basis of sequence similarity [25]. Some LOXs can co-oxidize Car [26,27,28,29,30,31]. Many LOX-related studies have examined the functions of LOXs in plant development and abiotic and biotic stresses [32]. However, the functional differences among the entire *LOX* gene family in plant species have remained unclear. 

Cucumber (*Cucumis sativus* L.), which belongs to the family Cucurbitaceae, is an economically important vegetable crop worldwide [33]. In the present study, a cucumber mutant that exhibits predominantly yellow-green leaf from spontaneous mutation was discovered. Whole genome resequencing and genetic mapping results showed non-synonymous mutations in the cucumber tandem *LOX* genes underlying the mutant phenotype. Consistent with the role of LOX in co-oxidation of Car and chloroplast destruction, abnormal chloroplast development was observed in leaf protoplasts of the mutant line. The present study reports the identification of the first mutant of the tandem *LOX* genes that affect Chl and Car biosynthesis in higher plants.

## 2. Result

### 2.1. ygl1 Mutant Has Reduced Chl Accumulation and Abnormal Chloroplast Development

*ygl1* mutant showing a yellow-green leaf phenotype was a spontaneous mutant isolated from inbred cucumber 1402. *ygl1* mutant was slightly smaller than the wild type throughout the developmental stage (Figure 1) and exhibited reduced levels of Chl a/b, as well as Car content (Table 1). Therefore, the net photosynthetic rate of *ygl1* mutant was significantly decreased compared with that of the wild type (Figure 1C). *ygl1* mutant leaves had a 72.7% and 73.2% reduction of Chl; however, only 9% and 10% reduction of Car levels were detected, compared with those in wild type at the 3- and 12-week-old stage, respectively. Thus, Car in the *ygl1* mutant slightly declined to that of the wild-type level. The Car/Pig ratio was 22.87% and 21.91% in *ygl1* mutant, but only 9.66% and 7.54% in wild type at the 3- and 12-week-old stage, respectively (Table 1). Thus, *ygl1* mutant exhibited severe yellow-green leaf coloration because of high rates of Car accumulation.

To investigate how the *ygl1* mutation affects chloroplast development, we compared the ultrastructures of plastids in the *ygl1* mutant and wild-type plants at three-week-old stages using TEM. The thylakoid lamellar structure in the *ygl1* mutant was abnormal, compared with those of wild-type leaves (Figure 1F, G), which might cause a reduction of the net photosynthetic rate (Figure 1C) of *ygl1* mutant, thereby explaining the distorted growth of the *ygl1* mutant to that of the wild type (Figure 1).

The phenotype of chloroplasts in the protoplasts isolated from three-week-old leaves of wild-type 1402 and the *ygl1* mutant was compared. Each protoplast isolated from the 1402 line contained more than 30 chloroplasts, but less than seven in the *ygl1* mutant (Figure 2A–C).

### 2.2. Identification of the Candidate Gene for ygl1

For the genetic analysis of the *ygl1* mutant, F_2_ populations were constructed from the crosses between the *ygl1* mutant and 1402. All F_1_ plants from the crosses displayed wild-type phenotype, and among the 156 F_2_ individuals, 34 displayed light green and 122 normal dark green, which fitted the 1:3 segregation ratio (green/yellow-green plants, $X^{2}<X_{0.05,1}^{2}$ = 3.84; *p* > 0.05; Figure 2D). These results suggested a single recessive gene underlying the mutated yellow-green phenotype.

Whole-genome resequencing was performed on the bulked genomic DNA samples from 33 yellow-green leaf plants and 33 normal plants from the F_2_ population (1402 and *ygl1*). Illumina high-throughput sequencing generated 28.31 G and 27.95 G clean bases from the 1402 pool and the *ygl1* pool, with a coverage of 93.74% and 93.79%, respectively. The effective sequencing depths for 1402 and *ygl1* were 85.73 times and 79.29 times the genome coverage, respectively, which guaranteed the accuracy of subsequent analysis.

Two paternal lines, namely, 1402 and *ygl1*, were resequenced because no reference genome sequences were available for these two cucumber lines. First, short reads from both parents were aligned with the 9930 genome to obtain two consensus sequences, which were used as reference sequences in the subsequent analysis. Second, reads obtained from two DNA-bulks (i.e., 1402 pool and *ygl1* pool) were aligned with consensus sequences to identify single-nucleotide polymorphisms (SNPs). To identify the genomic regions associated with yellow-green leaf, the proportion of SNP bases available between the 1402 pool and the *ygl1* pool was evaluated. The SNP-index was calculated for each SNP identified in the genome, and the sites with SNP-index less than 0.3 or greater than 0.7 in the 1402 pool and *ygl1* pool of this site were removed. The average SNP-index within a 1 Mb window size was computed using a 250 kb step increment. SNP-index graphs for the 1402 pool and *ygl1* pool were plotted by aligning an average SNP-index against the position of each sliding window in the 1402 reference genome (Figure 3A). The Δ(SNP-index) was calculated and plotted using the information from two graphs for 1402-bulk and *ygl1*-bulk (Figure 3A). Statistical confidence intervals of Δ(SNP-index) were calculated for all the SNP positions with given read depths. The chance that Δ(SNP-index) would become higher than 0.4778, as observed for the chromosomal region of 11.0–12.0 Mb on chromosome 4, was *p* < 0.05 under the null hypothesis.

A total of 36 putative ORFs (Open Reading Frames 1–36) were predicted in the region (File S1), and 10 *LOX* genes were present in tandem in the region. LOX isoenzymes have the capacity to co-oxidize Car [26]. Given that *ygl1* mutant exhibited severe yellow-green leaf coloration during photomorphogenesis because of high rates of Car accumulation, 10 *LOX* genes were selected as good candidates for *ygl1*. Because the target mutation conferred a recessive phenotype, homozygous SNPs in the *ygl1* pool in the 10 *LOX* genes were searched, and 36 SNPs (Table 2) were found. SNPs were displayed in all the coding sequences. Thus, non-synonymous SNPs in the F_2_ populations were examined, and four SNPs co-segregated with the yellow-green phenotype (Figure 3B). Among the four SNPs, SNP4G11124523 displaying a T to A transition between the wild type and *ygl1* pool caused a non-synonymous change within the gene *Csa4M286960* (F73Y); SNP4G11166969 displaying a G to T transition between the wild type and *ygl1* pool caused a non-synonymous change within the gene *Csa4M287550* (W112L); SNP4G 11182070 displaying an A to G transition between the wild type and *ygl1* pool caused a non-synonymous change within the gene *Csa4M288070* (E384G); and SNP4G11193182 displaying a C to T transition between the wild type and *ygl1* pool caused a non-synonymous change within the gene *Csa4M288080* (S203P) (Figure 3C). To infer the function of gene mutations, a comparison analysis was performed among the *LOX*s in cucumber and the isolated *LOX*s in *Oryza sativa*, *Physcomitrella patens*, *Vigna unguiculate*, and so on (Figure 3C). In *ygl1* mutant, the mutation led to an amino acid substitution of *Csa4M286960* (F73Y), *Csa4M287550* (W112L), *Csa4M288070* (E384G), and *Csa4M288080* (S203P) at a highly conserved residue. It indicated that the mutation sites in the four *LOXs* were diverse, compared with those of other species (Figure 3C), which implied that SNP mutation resulted in different amino acids, causing phenotypic changes. Moreover, the expression of four *LOX* genes significantly decreased in the *ygl1* mutant compared to the 1402 (Appendix A). Therefore, the combination of genome sequencing, segregation, and expression analysis indicated that four SNPs in four candidate *LOX* genes probably were the causative mutation of the yellow-green phenotype.

### 2.3. Abnormal Chloroplast Morphology in LOX Genes Knockdown Cucumber

Full-length cDNAs of the four *LOX* genes were isolated from cucumber via RT-PCR, based on the cucumber genomic sequence. *Csa4M286960* (GeneBank Accession number: XP_011653576.1), *Csa4M287550* (XP_011653578.1), *Csa4M288070* (XP_004142236.1), and *Csa4M288080* (XP_004142137.1) encoded 831, 821, 904, and 911 amino acids, respectively. The four *LOX* genes showed high similarity (Appendix A). Among the genes, Csa4M288070 and Csa4M288080 amino acids shared the highest similarity of 76%, and Csa4M288070 and Csa4M287550 shared the lowest similarity of 42%.

Phylogenetic analysis indicated that the four LOX genes were classified in the 13-LOX category into type II (Appendix A), according to the presence or absence of an N-terminal transit peptide of 102 *LOX* gene sequences from 17 plant species [34].

To further confirm the tandem four *LOX* gene function, transient RNAi was performed on cotyledons of the wild-type cucumber line 1402. The gene sequences of *Csa4M286960*, *Csa4M287550*, *Csa4M288070*, and *Csa4M288080* had high similarity. Thus, separately interfering with the expression of individual genes in this cluster would be difficult. Among the four genes, *Csa4M286960* was highly expressed in leaves (Appendix A). Therefore, we selected two target sites, *960-1:* 375 bp (referred to as “Ri1” hereafter) and *960-2:* 403 bp (referred to as “Ri2” hereafter) fragments of *Csa4M286960* for transient RNAi. The expression of *LOX* genes from the RNAi sample was reduced, compared with that in the control sample in Ri1 and Ri2, except for that of *Csa4M287550* in Ri1, which was slightly induced (Figure 4A). Accordingly, the proportion of abnormal protoplasts from the RNAi sample of Ri1 and Ri2 was 2.5 and 6.2 times, respectively, more than that of the control sample (Figure 4B). Compared with that of the control sample, the protoplasts in cotyledons with transient RNAi of *960-1* and *960-2* had fewer chloroplasts (Figure 4C–E) and had similar morphology to that of the chloroplasts in the yellow-green leaves of the *ygl1* mutant. The results indicated that the reduced expression of *960-1* might block chloroplast division in cucumber to a certain extent and reduce the expression of *960-2*, severely blocking chloroplast division. This finding supported that the *LOX* genes were the genes underlying the *ygl1* mutant.

VIGS was performed to test whether *960-1* and *960-2* silencing could cause the yellow-green phenotype. VIGS can serve as an alternative to stable transgenic plants to allow the characterization of gene functions in a wide range of plants. Therefore, VIGS silencing of *960-1* and *960-2* was performed on 1402. The results were consistent with the observation on transient RNAi silencing. PDS gene fragments (300 bp) amplified from mRNAs of cucumber leaves were inserted into TRV-RNA2 vectors. The resulting viruses (*cuPDS*-TRV) were inoculated in the cucumber cotyledons. All plants inoculated with *cuPDS*-TRV developed white leaves with highly uniform *PDS* knockout phenotype, indicating that the *PDS* gene had been silenced by *cuPDS*-TRV (Figure 5A–C).

Similarly, the plants were infected with *960-1*-TRV and *960-2*-TRV and the empty vector as a control. As the empty vector of TRV was restricted to the inoculated leaves of cucumber plants, and no silenced phenotypes appeared (Figure 5A–C), the growth of silenced plants was suppressed due to the loss of Chl. The silencing of *960-1*-TRV and *960-2*-TRV resulted in the yellow-green phenotype, and Chl was significantly reduced, compared with that of the control (Figure 5D). The number of chloroplasts in protoplasts isolated from the control was more than that in *cuPDS*-TRV, *960-1*-TRV, and *960-2*-TRV (Figure 5E). Thus, the *LOX* gene cluster that showed strong association with the yellow-green phenotype was proposed. Although the plants infected with *960-1*-TRV and *960-2*-TRV showed yellow-green phenotype, the effect was weaker than that of the *ygl1* mutant.

## 3. Discussion

LOXs are important dioxygenases in cellular organisms, and they contribute to plant developmental processes and environmental responses [35]. LOXs from several plants have been reported [32]. The *Arabidopsis* genome comprises six LOXs (i.e., *AtLOX1*–*AtLOX6*) [36]. *AtLOX1* is induced by pathogen attack, abscisic acid, and methyl jasmonate [37]; *AtLOX2* and *AtLOX6* are involved in jasmonic acid (JA) biosynthesis [38,39], and *AtLOX2* is essential for the formation of green leaf volatiles and five carbon volatiles [40]; *AtLOX3* and *AtLOX4* are essential for flower development [41]; *AtLOX5* plays an important role in lateral root development [42]. In tomato, *SlLOXA*, *SlLOXB*, and *SlLOXE* are upregulated during fruit ripening [43,44,45]; *SlLOXC* participates in production of flavor compounds resulting from fatty acids [45]; *SlLOXD* is involved in wound-induced JA biosynthesis, enhancing resistance against herbivores and pathogens [46]; *SlLOXF* enhances systemic resistance stimulated by *Pseudomonas putida* BTP1 [47]. Furthermore, in tobacco, *NaLOX2* is involved in green leaf volatiles (GLVs) biosynthesis [48,49]; and in potato, *StLOXH1* mediates the biosynthesis of volatile C6-aldehydes (GLVs) involved in defense [50]. The pepper *CaLOX1* functions in defense and cell death responses to microbial pathogens [51], and soybean *13-LOX* accomplishes a key role in the destruction of chloroplasts in senescing plants and might have a critical role in the NH2-terminal chloroplast transit peptide [52]. However, to our knowledge, no corresponding LOX mutant has been found and no *LOX* gene was characterized in cucumber. In the present study, the yellow-green leaf LOX mutant of cucumber was identified and functionally characterized. Whole genome resequencing of the wild-type parental line and pooled F_2_ progenies with mutated phenotype and genetic analysis were conducted, and the causative mutation to an interval with an enrichment of homozygous mutated SNPs was delimited. Mutations of the *LOX* gene cluster with a 9.2kb genomic interval were revealed to underlie the *ygl1* mutant.

LOXs, which are widely found in plants, fungi, and animals, are a large family of monomeric proteins with the capacity to co-oxidize Car [26,28,31]. Car and Chl are major photosynthetic pigments in plants, and all organisms are capable of photosynthesis. Car of higher plant chloroplasts function as collectors of light energy for photosynthesis and play an important structural role in the membrane stabilization of chloroplasts [3]. The yellow color of fruits and vegetables is attributed to Car and green to Chl [5]. The LOX mutant in the present study contains high levels of car pigments and produces a yellow-green leaf (Figure 1).

The content of photosynthetic pigments in plants is closely related to the variation of plant leaves [6,53]. In the present study, the Chl a and Chl b content of the mutant leaves was lower than that in the normal plants, indicating that the mutant *ygl1* belongs to the total Chl-type leaf color mutant. The ratio of total Chl to Car in the control was higher than that of the mutant. Therefore, the yellowing of the mutant *ygl1* leaf was caused by a significant decrease in Chl content. The decrease of Chl content could seriously affect photosynthesis. A decrease in plant photosynthetic pigments hinders photochemical conversion efficiency [54]. This finding is consistent with the observation that the net photosynthetic rate of yellow-green leaf mutant *ygl1* was 43% lower than that of the control. The decrease of photosynthetic pigment content in the mutant may affect the net photosynthetic efficiency of the plant.

A 13-LOX enzyme accumulated in the plastid envelope and catalyzed the deoxygenation of unsaturated membrane fatty acids, leading to a selective destruction of the chloroplast and the release of stromal constituents during plant leaf senescence [52]. In our study, the *ygl1* leaf showed reduced Chl accumulation and abnormal chloroplast development upon 13-LOX inhibition. This result is consistent with the observation that the *13-LOX* gene *LOX2* is essential in the formation of GLVs and five-carbon volatiles [40]. LOX is responsible for the co-oxidation of Car [26]. In the present study, the cucumber LOX mutant displayed yellow-green leaves. Large and few chloroplasts were observed in the leaf protoplasts of the LOX mutant (Figure 2B). Furthermore, the chloroplast phenotype in the *ygl1* mutant was very similar to that in LOX transient RNAi and VIGS plants, supporting that the mutation in LOXs underlies chloroplast defects. This evidence suggests that the mutations in LOXs and cucumber *ygl1* might affect Car biosynthesis, and chloroplast development may act in similar modes. The relative highly enriched Car phenotype in *ygl1* was the result of tandem *LOX* gene mutation (Table 1). This finding provides overwhelming evidence that LOX isoenzymes have the capacity to co-oxidize Car [26]. Detailed annotation of the 9.2 kb region harboring the *LOX* locus identified a cluster of 10 *LOX* gene analogs. Four LOXs were in the region delimited by SNPs (i.e., SNP4G11124523, SNP4G11166969, SNP4G 11182070, and SNP4G11193182) and were possible candidates of *ygl1* mutation.

Carotenoids serve as antenna in the light-harvesting proteins of photosynthesis, which absorb sunlight in the blue and green parts of the solar spectrum and transfer the energy to nearby Chl molecules for photochemical conversion [55]; they also regulate light-energy conversion in photosynthesis and protect plants from reactive oxygen species and coloration [26,56]. In humans, provitamin A Car (i.e., α-carotene, β-carotene, γ-carotene, and the xanthophyll β-cryptoxanthin) are best known for being converted enzymatically into vitamin A; diseases resulting from vitamin A deficiency remain among the most significant nutritional challenges worldwide [26]. Car are widely consumed, and their consumption is a modifiable health behavior (via diets or supplements), resulting in health benefits for chronic disease prevention. LOX inhibition could block the degradation of most Car in plants, which can be very significant for public health [57]. In the case of golden rice, downregulation of lipoxygenase enzyme activity could reduce degradation of Car, which reduces huge postharvest and economic losses of biofortified rice seeds during storage [27]. In the present work, knockdown *LOX*s in cucumber retained the high enrichment of Car, and this might shed light on the tools needed to reduce Car losses in plants.

In plants, the occurrence of disease-resistant genes in clusters is critical for generating diversity of resistance specificities because the tandem arrays support high rates of gene conversion and illegitimate recombination [58]. Complex histories of transposon insertions, translocations, and gene duplications and rearrangements have also contributed to the formation of gene clusters [59]. Chromosomal localization and genome distribution of *CsLOX* genes have revealed that tandem duplication and/or polyploidy duplication based on the cucumber genome annotation database might contribute to the expansion of *CsLOX* genes. On chromosomes 2 and 4, which are 18 and 20 *CsLOX*, respectively (Appendix A), a cluster with a relatively high density was observed. These *CsLOX* members have a high sequence similarity with each other. Exploration of the detailed mechanism of tandem cluster of *LOX* genes in plant Car metabolism is of significant interest.

## 4. Materials and Methods

### 4.1. Plant Materials

Cucumber (*Cucumis sativus* L.) *ygl1* (yellow-green leaf) mutant was isolated from a South China-type cucumber inbred line 1402. The *ygl1* is a spontaneous natural mutant, after three generations, the inherited stably *ygl1* mutant was used for reciprocally crossing with 1402 to construct the F_2_ segregating population. All plants were grown in a greenhouse in the science and technology building of Southwest University, Chongqing, China. 

### 4.2. Net Photosynthetic Rate

The net photosynthetic rates (Pn) of wild type (WT) and *ygl1* leaves were measured with CB-1101 photosynthetic system (Siaidi, Beijing, China). The instrument setting parameters are as follows: The light intensity was set to 1000 μmol·m^−2^·S^−1^ for measurement. Pn of the plants was determined by setting the chamber temperature to 10 °C, and the constant leaf area was 6.5 cm^2^. Measurements were carried out in clear and cloudless weather from 9:00 am to 11:00 am. After 30 days of sowing, uniform cucumber leaves were selected from WT and *ygl1*, and six plants were tested for WT and *ygl1*, respectively, and each assay was repeated 10 times.

### 4.3. Transmission Electron Microscopy (TEM) Analysis

Wild-type and *ygl1* mutant leaf samples were harvested from 3-week-old plants grown in a greenhouse. The leaves were cut with a sharp blade into small pieces of approximately 1 mm^2^. The following procedures were performed as described previously [60] with slight modifications. In brief, leaf sections were fixed in a solution of 2% glutaraldehyde (pH = 7.4) and further fixed in 2% OsO_4_. Tissues were stained with uranyl acetate, dehydrated in ethanol, and embedded in ethoxyline resin prior to thin sectioning. Leica UC6 ultrathin slicer was used for slicing, and uranyl acetate was used for staining. TEM was used to observe and photograph by HITACHI H-7650 (Tokyo, Japan).

### 4.4. Chloroplast Phenotype Analysis

Isolation of cucumber leaf protoplast and chloroplast phenotype analysis were performed according to the reported method [61]. For each biological replicate, 50 protoplasts were counted, and the protoplast was considered abnormal if it harbored less than 15 chloroplasts.

### 4.5. Whole Genome Resequencing

Genomic DNA was extracted from fresh leaves by using standardized CTAB protocol [62]. The DNA of 33 plants showing a mutant phenotype in the F_2_ population was mixed equally to construct the *ygl1* pool. The DNA of 33 plants showing normal green leaves in the F_2_ population was mixed equally to construct the 1402 pool. Two DNA pools were prepared by mixing equimolar concentration of DNA samples. For Illumina paired-end sequencing, the sequencing library with an insert size of 500 bp was constructed according to the standard protocol. Then, the library with a read length of 100 bp was sequenced using Illumina Hi-Seq 2500 sequencer. The sequencing generated 28.31 G clean base for 1402 and 27.95 G for the *ygl* pool. In both cases, more than 96% of bases possessed an Illumina Phred quality score higher than 20.

### 4.6. Sequence Data Analysis

The short reads of the 1402 and *ygl1* pools were aligned against the reference genome of cucumber inbred line *Cucumis sativus* L. var. sativus cv. 9930 [63] using BWA-MEM [64]. The output of paired-end alignment files was processed with SAMtools [65] with default parameters to generate consensus genome sequences. The SNP (single nucleotide polymorphism)-index was carried out by directly comparing the consensus sequences of 1402 and ygl1 pools, base-for-base [15]. The index is equal to 1 when all short reads differ from the reference genome, and is equal to 0 when all short reads are identical to the reference genome. In homozygous crops, changes in the SNP-index and Δ(SNP-index) in the offspring pool are usually caused by the exchange and recombination of homologous chromosomes of the genome of one of the parents taken as the reference sequence.

Using the Δ(SNP-index) for sliding window analysis, changes in the SNP-index and Δ(SNP-index) of the SNP sites within the candidate interval are consistent with the recombination section, and the candidate section and the different phenotypes correspond to the sign of the Δ(SNP-index) window.

Given that the SNP-index was based on a reference genome rather than the genome of the parental line, the origin of each SNP or the linkage around its neighboring region could not be estimated. Positive or negative values of the Δ(SNP-index) did not correspond to the phenotype of the parental lines. Δ(SNP-index) was used as the main parameter to identify the target phenotype. The 95% and 99% confidence intervals of the Δ(SNP-index) under the null hypothesis of no QTLs (Quantitative Trait Locus) were calculated following the method described previously [66].

### 4.7. Sequence Alignments and Phylogenetic Analysis

The genome sequences and annotation of cucumber inbred line 9930 are accessible in the GenBank under the accession ACHR00000000. The sequence alignment was performed with ClustalW, and the phylogenetic tree was constructed via neighbor-joining using MEGA5.05.

### 4.8. Development of PCR-Based SNP Markers

SNPs were typed by amplifying an SNP-containing DNA fragment with two flanking primers. All the PCR reactions were performed in a PCR Thermal Cycler SP (Takara). PCR products were visualized using gel electrophoresis on 2% (*w*/*v*) agarose. The primers are summarized in Appendix A.

### 4.9. Transient Gene Silencing in Cucumber Cotyledons

Transient RNAi system was performed on cucumber cotyledons according to a previous protocol [67]. To construct the RNA interference vector, 375 bp (Ri1) and 403 bp (Ri2) fragments were amplified from the *Csa4M286960* coding sequence using primers with a 5′-attB1 extension forward primer and a 5′-attB2 extension reverse primer (Appendix A, 5′-attB1 and 5′-attB2 extensions are underlined). A recombination reaction between the PCR product and the pHellsgate 2 vector (Invitrogen, USA) was carried out using BP clonase (Invitrogen) according to the manufacturer’s instructions. Constructs were separately transformed into *Agrobacterium tumefaciens* strain LBA4404. Afterwards, the agrobacterium suspension was infiltrated into cotyledons of wild-type line 1402 seedlings. Five days post-injection infiltrated, the cotyledons were cut off for chloroplast observation and quantitative real-time PCR analysis (the primer sequences are shown in Appendix A). Three biological replicates were conducted in this experiment.

### 4.10. Chl Content and SPAD Assay

Chl content was measured via the Lichtenthaler method [68]. Leaf tissues were ground in liquid nitrogen and extracted with 8 mL of 80% (*v*/*v*) acetone. Absorption spectra were detected at 663, 646, and 470 nm. Chl was computed using the equation as follows: Chl concentration (mg/L) = C_a_= 12.21A_663_ – 2.81A_646,_ C_b_ = 20.13A_646_ – 5.03A_663_, C_T_ = C_a_ + C_b_ = 17.32A_646_ – 7.18A_663_, C_x.c_ = (1000A_470_ – 3.27 C_a_ – 104 C_b_)/229, where *A* is the absorbance at a specified wavelength.

Chl meter values (SPAD values) were taken with the Minolta SPAD-502 Chl meter.

### 4.11. Plasmid Construction and Agro-infiltration

pTRV1 and pTRV2 virus-induced gene silencing (VIGS) vectors were used as described previously [69].

pTRV2-*CuPDS* construction: The sequence of cucumber PDS genes (EF159942) was amplified from mRNA samples from cucumber leaves using primer pair cuPDS31(+) (corresponding to nt positions 31–50) and cuPDS900(−) (complementary to nt positions 881–900) for a PDS gene (EF159942) [70]. The resulting product was cloned into pTRV2 to form pTRV2-*CuPDS*.

pTRV2-*960-1* construction: The 375 bp fragment of the Csa4M286960 gene was PCR-amplified from cucumber cDNA by using primers 960TRV-1FW(+) (corresponding to nt positions −3–17) and 960TRV-1RV(−) (complementary to nt positions 353–372). The resulting PCR product was cloned into pTRV2 to form pTRV2-*960-1*.

pTRV2-*960-2* construction: The 403 bp fragment of the 960 gene was PCR amplified from cucumber cDNA by using primers 960TRV-2FW (+) (corresponding to nt positions 427–446) and 960TRV-2RV(−) (complementary to nt positions 810–829). The resulting PCR product was cloned into pTRV2 to form pTRV2-*960-2*.

The primers are summarized in Appendix A, and the added restriction enzymes site of primers is underlined. All the PCR products were cloned into the pMD18-T vector (TaKaRa, Dalian, China) and sequenced, then digested with restriction enzymes *Bam* HI and *Xho* I. The resulting product was inserted into the *Bam* HI and *Xho* I sites of the vector pTRV2 to yield the above construction.

Plant infiltration was performed as described previously [71]. *Agrobacterium* strain GV3101 containing pTRV1 or pTRV2 and its derivatives was used for VIGS experiments. *Agrobacterium* strain GV3101 containing TRV-VIGS vectors was grown at 28 °C in LB medium containing 10 mm MES and 20 mm acetosyringe with appropriate antibiotics. After 24 h, *Agrobacterium* cells were harvested and resuspended in the *Agrobacterium* infiltration buffer (10 mm MgCl_2_, 10 mm MES, pH 5.6, 150 mm acetosyringone) to a final OD_600_ of 1.0 (for both pTRV1 and pTRV2 or its derivatives) and shaken for 4–6 h at room temperature before infiltration. For cucumber leaf infiltration, each *Agrobacterium* strain containing pTRV1 and pTRV2 or its derivative vectors was mixed in a 1:1 ratio and infiltrated into the cucumber leaves with a 1 mL needleless syringe.

### 4.12. Statistical Analysis

Data was analyzed using variance by SAS software (version 8.0, SAS Institute, NC, USA), and statistical differences were compared using Fisher’s least significant difference (LSD) test.

## Figures and Tables

**Figure 1 ijms-20-03102-f001:**
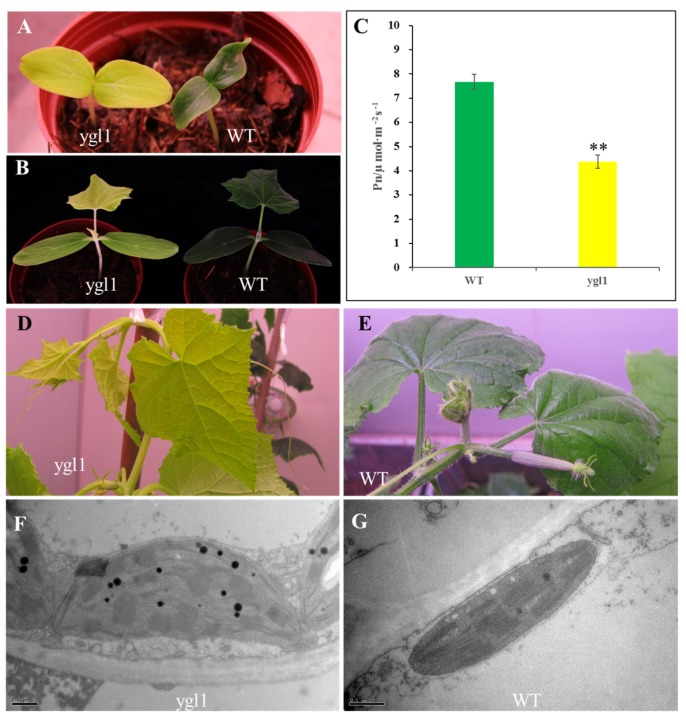
Phenotypic characterization of the cucumber *ygl1* mutants. (**A**) One-week-old plants. (**B**) Three-week-old plants. (**C**) Net photosynthesis rate of leaves in *ygl1* and wild type (WT) “1402” in cucumber. *n* = 10, ** stands for significant difference (*p* < 0.01). (**D**,**E**) Twelve-week-old plants. (**F**) Chloroplasts of three-week-old *ygl1* mutant have abnormal thylakoid lamellar structure to those of (**G**) the wild type with well-ordered thylakoid lamellar structure in three-week-old plants. Bar equals 0.5 μm.

**Figure 2 ijms-20-03102-f002:**
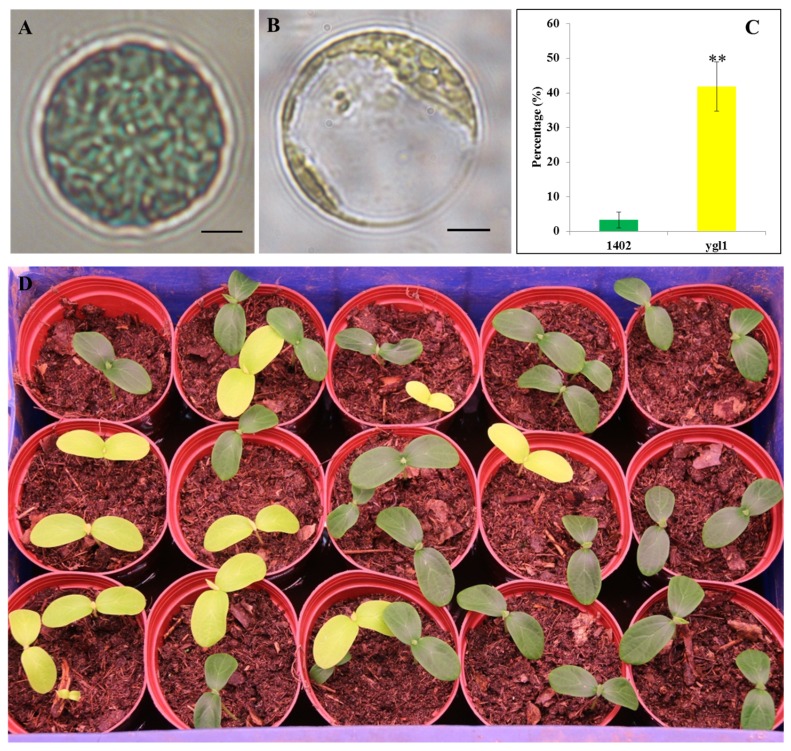
Chloroplast morphology and segregation of F_2_ populations in wild type 1402 and *ygl1* mutant. (**A**,**B**) Chloroplast morphology of the protoplast isolated from wild type 1402 and *ygl1* mutant leaves. Scale bar = 10 μm. (**C**) Quantitative proportion of protoplasts with abnormal chloroplast number in cotyledons infiltrated with agrobacterium. For each biological replicate, 50 protoplasts were counted and an abnormal protoplast harbored less than 15 chloroplasts. ** stands for significant difference (*p* < 0.01). (**D**) Phenotypic segregation of part F_2_ populations in wild type 1402 and *ygl1* mutant.

**Figure 3 ijms-20-03102-f003:**
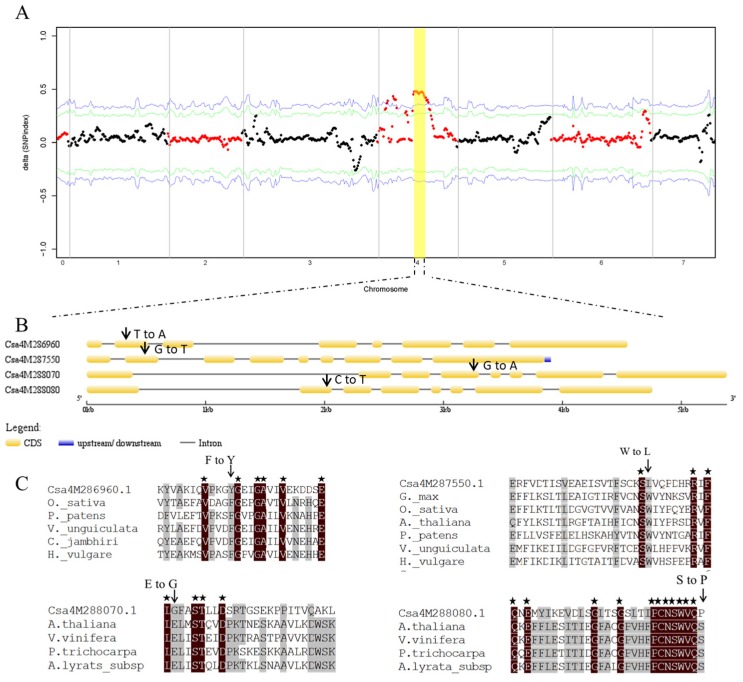
Identification of the gene conferring the yellow-green phenotype. (**A**) Δ(SNP-index) distribution of the 1402 and *ygl1* pools. The *x* axis represents the position of seven chromosomes, and the *y* axis represents the Δ(SNP-index). The arrow and yellow rectangle indicate the region of chromosome 4 with a Δ(SNP-index) of approximately 0.5. The statistical confidence interval under the null hypothesis of no QTLs (Quantitative Trait Locus) was indicated as green (*p* < 0.05) and blue (*p* < 0.01). (**B**) Gene structure of *Csa4M286960*, *Csa4M287550*, *Csa4M288070*, and *Csa4M288080*. The black vertical line represents the mutations. (**C**) Alignment of the *LOX* genes from diverse species with *Csa4M286960*, *Csa4M287550*, *Csa4M288070*, and *Csa4M288080* homologs. Amino acid residues displaying >50% identity or similarity between the homologs are shaded black or gray.

**Figure 4 ijms-20-03102-f004:**
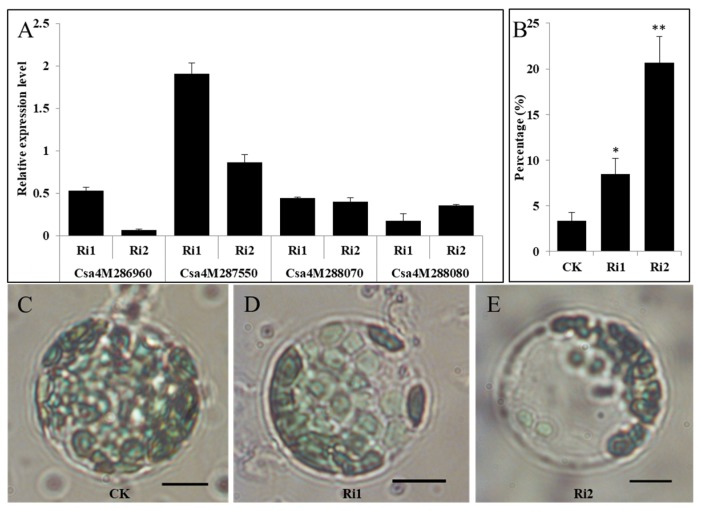
RNAi transient silencing of tandem *13-lipoxygenase* genes in cucumber cotyledons. (**A**) Transient gene expression of tandem *13-lipoxygenase* genes in cotyledons infiltrated with agrobacterium. The expressions of each sample infiltrated with RNAi (Ri1 and Ri2) were compared with that of the sample infiltrated with empty vector (CK) after normalization to the cucumber ubiquitin gene and presented as relative expression. (**B**) Quantitative proportion of protoplasts with abnormal chloroplast number in cotyledons infiltrated with agrobacterium. For each biological replicate, 50 protoplasts were counted, and an abnormal protoplast harbored less than 15 chloroplasts. Error bar indicates standard error. * stands for significant difference (*p* < 0.05), ** stands for significant difference (*p* < 0.01). (**C**–**E**) Chloroplast morphology of the representative protoplasts isolated from CK and Ri. Scale bar = 10 μm.

**Figure 5 ijms-20-03102-f005:**
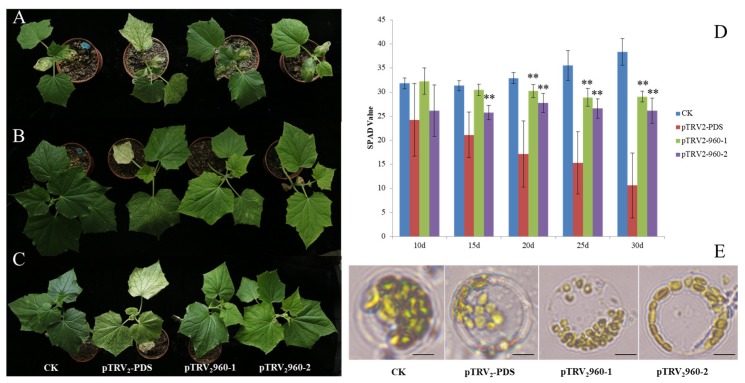
Virus-induced gene silencing in plants infected with TRV vectors of tandem 13-lipoxygenase genes in cucumber leaves. Cucumber leaf morphology co-infected with empty TRV_2_, pTRV_2_-*PDS*, pTRV_2_-*960-1*, and pTRV_2_-*960-2* after eight days post-inoculation (dpi) (**A**), 16 dpi (**B**), and 24 dpi (**C**). (**D**) Chl meter (SPAD) value of cucumber leaves co-infected with empty TRV2, pTRV2-PDS, pTRV2-*960-1*, and pTRV2-*960-2* at different dpi. (**E**) Chloroplast morphology of the representative protoplasts isolated from cucumber leaves, co-infected with empty TRV_2,_ pTRV_2_-PDS, pTRV_2_-*960-1*, and pTRV_2_-*960-2*. Scale bar = 10 μm. ** stands for significant difference (*p* < 0.01).

**Table 1 ijms-20-03102-t001:** Pigment (Pig) contents in leaves of wild-type and *ygl1* mutant, in mg/g fresh weight.

Growth Stage	Genotype	Chl a	Chl b	Car	Car /Pig Percent
3 weeks old	Wild type	2.39 ± 0.20	1.31 ± 0.14	0.39 ± 0.04	9.66 ± 2.60
	*ygl1*	0.76 ± 0.03	0.25 ± 0.02	0.30 ± 0.01	22.87 ± 1.25
12 weeks old	Wild type	2.52 ± 0.15	1.17 ± 0.07	0.30 ± 0.02	7.54 ± 0.75
	*ygl1*	0.76 ± 0.06	0.23 ± 0.05	0.27 ± 0.01	21.91 ± 1.39

Chl and Car were measured in acetone extracts from third leaf of different growth stages from top. Values shown are the mean SD from three independent determinations.

**Table 2 ijms-20-03102-t002:** Single nucleotide polymorphisms (SNPs) distribution in the region of 10 predicted *LOX* genes present in tandem in the cucumber genome chromosome 4.

Position	Ref	ygl1	Gene	Translate	aa	Annotation
11124523	A	T	*Csa4M286960.1*	TTT<->TAT	F<->Y	Lipoxygenase
11126210	T	G	*Csa4M286960.1*	CCT<->CCG	P<->P	Lipoxygenase
11126739	A	G	*Csa4M286960.1*	TTA<->TTG	L<->L	Lipoxygenase
11135803	T	C	*Csa4M286980.1*	GAC<->GAT	D<->D	Lipoxygenase
11135804	A	G	*Csa4M286980.1*	GAT<->AAT	D<->N	Lipoxygenase
11135807	T	G	*Csa4M286980.1*	TGA<->GGA	*<->G	Lipoxygenase
11141217	A	G	*Csa4M286990.1*	ACG<->ACA	T<->T	Lipoxygenase
11144638	C	G	*Csa4M287010.1*	GCG<->GGG	A<->G	Lipoxygenase
11166969	T	G	*Csa4M287550.1*	TGG<->TTG	W<->L	Lipoxygenase
11167955	A	C	*Csa4M287550.1*	GCT<->GAT	A<->D	Lipoxygenase
11168050	C	G	*Csa4M287550.1*	GTT<->CTT	V<->L	Lipoxygenase
11168212	T	C	*Csa4M287550.1*	TAT<->TAC	Y<->Y	Lipoxygenase
11168253	T	C	*Csa4M287550.1*	GTT<->GCT	V<->A	Lipoxygenase
11173198	A	G	*Csa4M287570.1*	ACA<->ATA	T<->I	Unknown protein
11178904	A	G	*Csa4M288070.1*	TGG<->TAG	W<->*	Lipoxygenase
11179163	T	G	*Csa4M288070.1*	TTG<->TTT	L<->F	Lipoxygenase
11181197	T	C	*Csa4M288070.1*	TCG<->TTG	S<->L	Lipoxygenase
11181330	A	G	*Csa4M288070.1*	CAA<->CAG	Q<->Q	Lipoxygenase
11181348	A	G	*Csa4M288070.1*	CAG<->CAA	Q<->Q	Lipoxygenase
11181801	T	G	*Csa4M288070.1*	TCG<->TCT	S<->S	Lipoxygenase
11181936	T	C	*Csa4M288070.1*	GAT<->GAC	D<->D	Lipoxygenase
11181988	A	G	*Csa4M288070.1*	GAT<->AAT	D<->N	Lipoxygenase
11182010	T	C	*Csa4M288070.1*	CCT<->CTT	P<->L	Lipoxygenase
11182070	A	G	*Csa4M288070.1*	GAA<->GGA	E<->G	Lipoxygenase
11182092	A	G	*Csa4M288070.1*	CAG<->CAA	Q<->Q	Lipoxygenase
11182243	T	C	*Csa4M288070.1*	TTT<->TTC	F<->F	Lipoxygenase
11190230	T	C	*Csa4M288080.1*	ATC<->ACC	I<->T	Lipoxygenase
11192106	T	C	*Csa4M288080.1*	ACT<->ATT	T<->I	Lipoxygenase
11192300	A	G	*Csa4M288080.1*	GTA<->GTG	V<->V	Lipoxygenase
11192333	A	G	*Csa4M288080.1*	CAA<->CAG	Q<->Q	Lipoxygenase
11192417	T	C	*Csa4M288080.1*	CTC<->CTT	L<->L	Lipoxygenase
11193182	T	C	*Csa4M288080.1*	CCT<->TCT	P<->S	Lipoxygenase
11203061	A	G	*Csa4M288110.1*	GCG<->GCA	A<->A	Lipoxygenase
11203109	T	C	*Csa4M288110.1*	TCC<->TCT	S<->S	Lipoxygenase
11203161	A	T	*Csa4M288110.1*	TAT<->AAT	Y<->N	Lipoxygenase
11216771	A	G	*Csa4M288610.1*	ACG<->ACA	T<->T	Lipoxygenase

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
