# Peer review of "Tandem 13-Lipoxygenase Genes in a Cluster Confers Yellow-Green Leaf in Cucumber"

_ijms, 2019, doi:10.3390/ijms20123102_

Reviewer 1 Report

I have reviewed the ms titled "Tandem 13-lipoxygenase genes in a cluster confers yellow-green leaf in cucumber". In the ms, authors discovered a yellow-green leaf mutant in cucumber and identified LOX gene using genetic mapping. However, some questions need to be figured out before published.

For paper writing, the ms must be improved, some places were really hard to understand.

How do authors obtained the ygl1 mutant from 1402, natural mutant or induced? and how many generations after derived from 1402? how to reproduce after derived from 1402?

In methods, several places of citations were needed, such as TEM analysis. For CB-1101 photosynthetic system, more detail information was needed.

The resolution of figures was needed to be improved.

Row 99-102, authors mentioned that "DNA of … was mixed equally to construct the … pool", but how? need to be more detail.

Row 110-111, what authors mean by "genome sequence consensus"?

Row 112-127, the description was so confusing, authors need to reorganize this method. such as, what is a "mutant SNP", and what is "the number of reads corresponding to SNP"

Row 244-245, authors mentioned "the sites with SNP-index less than 0.3 or greater than 0.7 in 1402-pool and ygl1-pools of this site were removed", but why?

Row 251-252, authors mentioned "10.6-11.5 Mb" as the association region, but as authors mentioned in above the sliding window is 1 Mb, so I think if one window was identified as an association, at least the region should be 1 Mb, right? Addition, what is the null hypothesis?

In figure 3B, why are the four genes overlap in the genomic region?

In row 276-280, authors wrote "To infer the function of gene mutations, a comparison analysis was performed among the LOXs in cucumber and the reported LOXs in Oryza sativa, Physcomitrella patens, and Vigna unguiculata. The result indicated that the mutation sites in the four LOXs were diverse compared with that of other species (Fig. 3C), which implied that SNP mutation resulted in different amino acids, causing phenotypic changes.", but what are the "result"? after comparison, authors not give any result. Even the authors clearly showed the result, I don't think the result could "indicated the four SNPs in the four LOX genes were causative mutations". This just provided four candidates.

In 303-305, What were authors want to say for "Accordingly, the proportion of protoplasts with reduced chloroplast number from RNAi sample of Ri1 and Ri2 was 2.5 and 6.2 times, respectively, more than that of the control sample (Fig. 4B)"?

In candidate gene identification, authors just considered the genes with non-synonymous SNP as candidates, while in function validation, the authors used the expression. And through all MS, authors even not compared the candidate expression between mutant and wild type.

In reference 40, what is the "[34]" means?

Author Response

We are very thankful for a thorough review and for the helpful comments regarding our manuscript entitled “Tandem 13-lipoxygenase genes in a cluster confers yellow green leaf in cucumber”. We have fully revised the manuscript based on your comments. Changes made to the text are "Track Changes" so that you can be easily identified.

Listed below are the point-by-point responses to your suggestions and comments.

How do authors obtained the ygl1 mutant from 1402, natural mutant or induced? and how many generations after derived from 1402? how to reproduce after derived from 1402?

Response(R)The ygl1 is a spontaneous natural mutant, after three generations selfing, the inherited stably ygl1 mutant was used for reciprocally crossing with 1402 to construct the F2 segregating population.

In methods, several places of citations were needed, such as TEM analysis. For CB-1101 photosynthetic system, more detail information was needed.

RThanks, it has been revised as suggested. Changes made to the text regarding this are in lines 83, 86 to 87 and 92.

The resolution of figures was needed to be improved.

ROK, we have uploaded high resolution image separately.

Row 99-102, authors mentioned that "DNA of … was mixed equally to construct the … pool", but how? need to be more detail.

RThank you for the suggestion. We have added more detail. Changes made to the text regarding this are in lines 109 to 110.

Row 110-111, what authors mean by "genome sequence consensus"?

Row 112-127, the description was so confusing, authors need to reorganize this method. such as, what is a "mutant SNP", and what is "the number of reads corresponding to SNP"

RSorry for that, the genome sequence consensus should be consensus genome sequences. We have corrected it. The consensus genome sequences means the short reads of 1402 and ygl1 pool assembled sequence according to the reference genome of cucumber inbred line 9930 (Huang et al. 2009; Li et al. 2011). The "mutant SNP" means the SNPs linked to the mutant phenotype, and "the number of reads corresponding to SNP" means all the SNPs with sequence reads, the detail description is shown in the paper in 2012 (Abe, A.; Kosugi, S.; Yoshida, K.; Natsume, S.; Takagi, H.; Kanzaki, H.; Matsumura, H.; Yoshida, K.; Mitsuoka, C.; Tamiru, M.; Innan, H.; Cano, L.; Kamoun, S.; Terauchi, R., Genome sequencing reveals agronomically important loci in rice using MutMap. Nat Biotechnol 2012, 30, (2), 174-8.).

We have revised this part and deleted the confusing statement, changes made to the text regarding this are in lines 119 to 120.

Row 244-245, authors mentioned "the sites with SNP-index less than 0.3 or greater than 0.7 in 1402-pool and ygl1-pools of this site were removed", but why?

RThe SNP-index less than 0.3 or greater than 0.7 in both two pools means the Δ (SNP-index) ranged from -0.3 to 0.3. These SNPs are useless for identifying the candidate region according the 1:3 segregation ratio. So we removed these SNP in the future analysis. We refer to the following methods described: Takagi H, Abe A, Yoshida K, Kosugi S, Natsume S, Mitsuoka C, et al. QTLseq: rapid mapping of quantitative trait loci in rice by whole genome resequencing of DNA from two bulked populations. The Plant Journal. 2013;74(1):174-83.

Row 251-252, authors mentioned "10.6-11.5 Mb" as the association region, but as authors mentioned in above the sliding window is 1 Mb, so I think if one window was identified as an association, at least the region should be 1 Mb, right? Addition, what is the null hypothesis?

RThank you for pointing it out, we have reanalyzed the data and revised accordingly, and the analyses yielded the identical result as before. Null hypothesis is explained in the paper (Figure S6) ” QTLseq: rapid mapping of quantitative trait loci in rice by whole genome resequencing of DNA from two bulked populations” published in The Plant Journal. Changes made to the text regarding this are in lines 133 to 134. Changes made to the text regarding the association region is in line 279 and the Figure 3A and figure legend.

In figure 3B, why are the four genes overlap in the genomic region?

R: The four genes are not overlapped, figure 3B only describe the gene structure of Csa4M286960, Csa4M287550, Csa4M288070, and Csa4M288080 respectively. The four genes in the chromosome are as followsCsa4G286960  Chr4 : 11124202 .. 11128409 (+), Csa4G287550     Chr4 : 11165221 .. 11168458 (+), Csa4G288070     Chr4 : 11178840 .. 11183277 (+), Csa4G288080     Chr4 : 11190142 .. 11193283 (+)

In row 276-280, authors wrote "To infer the function of gene mutations, a comparison analysis was performed among the LOXs in cucumber and the reported LOXs in Oryza sativa, Physcomitrella patens, and Vigna unguiculata. The result indicated that the mutation sites in the four LOXs were diverse compared with that of other species (Fig. 3C), which implied that SNP mutation resulted in different amino acids, causing phenotypic changes.", but what are the "result"? after comparison, authors not give any result. Even the authors clearly showed the result, I don't think the result could "indicated the four SNPs in the four LOX genes were causative mutations". This just provided four candidates.

RThank you for pointing it out, we have added more details in the "result" and revised it accordingly. Changes made to the text regarding this are on lines 321 to 328.

In 303-305, What were authors want to say for "Accordingly, the proportion of protoplasts with reduced chloroplast number from RNAi sample of Ri1 and Ri2 was 2.5 and 6.2 times, respectively, more than that of the control sample (Fig. 4B)"?

RIt should be “Accordingly, the proportion of abnormal protoplasts from RNAi sample of Ri1 and Ri2 was 2.5 and 6.2 times, respectively, more than that of the control sample”. We have revised accordingly.

In candidate gene identification, authors just considered the genes with non-synonymous SNP as candidates, while in function validation, the authors used the expression. And through all MS, authors even not compared the candidate expression between mutant and wild type.

RThank you for pointing it out. Actually, we have compared the candidate genes expression between 1402 and ygl1. This is added in the figure S1 and add it to the text in lines 326 to 327. We have revised accordingly.

In reference 40, what is the "[34]" means?

RSorry for this mistake. We have deleted it.

Reviewer 2 Report

Ding et al isolate a spontaneous mutant in cucumber inbred line 1402 with yellow-green leaves, yg1. They demonstrate that this phenotype causes reduced chlorophyll content and increased carotenoid retention in yg1 leaves, and find four tandemly duplicated LOX genes have nonsynonymous changes (at least one of which,  Csa4M286960, is highly expressed in leaves), and RNA knockdown of these duplicates phenocopies the yg1 phenotype to a lesser extent.

Overall the paper is very clearly written and easy to follow. The methodology is sound and sufficient to demonstrate the provenance of this mutant. I have only a few minor comments to increase the clarity of the manuscript, below.

Line 203-204. Figure 1F & G, caption. It is unclear to me what this part of the caption means. Please rephrase.

Line 269. " and four SNPs were co-segregated with the yellow-green phenotype"

Should be "and four SNPs co-segregated with the yellow-green phenotype"

Line 278-279.Can you rephrase? "The mutation sites in the four LOXs were typically conserved", perhaps.

Line 285. Knockdown is misspelled.

Line 364. "corresponding LOX mutant has been found and LOX gene was characterized in cucumber.". Should be "corresponding LOX mutant has been found and no LOX gene has been characterized in cucumber."

Line 404. Can you expand on what you mean by carotenoids serving as antennae?

Line 418. Can you expand slightly in the discussion - how might a spontaneous mutant acquire so many changes within a single gene group? I think you were hinting at this with the 'illegitimate recombination' (do you mean, ectopic crossovers, or intergenic gene conversion?), so you might expand there.

Author Response

We are very thankful for a thorough review and for the helpful comments regarding our manuscript entitled “Tandem 13-lipoxygenase genes in a cluster confers yellow green leaf in cucumber”. We have fully revised the manuscript based on your comments. Changes made to the text are "Track Changes" so that you can be easily identified.

Listed below are the point-by-point responses to your suggestions and comments.

Line 203-204. Figure 1F & G, caption. It is unclear to me what this part of the caption means. Please rephrase.

Response (R)Thank you for the suggestion. We have revised it in the figure 1 F & G, and revised it accordingly in the text on line 235.

Line 269. " and four SNPs were co-segregated with the yellow-green phenotype"

Should be "and four SNPs co-segregated with the yellow-green phenotype"

RThanks for you point out. We have revised accordingly.

Line 278-279.Can you rephrase? "The mutation sites in the four LOXs were typically conserved", perhaps.

RThank you for the suggestion. We have revised accordingly. Changes made to the text regarding this are on lines 321 to 323

Line 285. Knockdown is misspelled. Line 364. "corresponding LOX mutant has been found and LOX gene was characterized in cucumber.". Should be "corresponding LOX mutant has been found and no LOX gene has been characterized in cucumber."

RThanks for you point out. We have revised accordingly. 

Line 404. Can you expand on what you mean by carotenoids serving as antennae?

RCarotenoids are indispensable in photosynthetic energy conversion, where they function as light harvesters and photoprotectors. They exert their light harvesting function by absorbing sunlight in the blue and green parts of the solar spectrum and transferring the energy to nearby Chl molecules for photochemical conversion. We have added more detail and reference in the text lines 465 to 466 to explain it.

Line 418. Can you expand slightly in the discussion - how might a spontaneous mutant acquire so many changes within a single gene group? I think you were hinting at this with the 'illegitimate recombination' (do you mean, ectopic crossovers, or intergenic gene conversion?), so you might expand there.

RThank you for your kind comments. It is hard to discuss. We are not sure on this question of how might a spontaneous mutant acquire so many changes within a single gene group. We need more investigation to explore detailed mechanism of tandem cluster of LOX genes in plant Car metabolism. This will be done in our follow-up experiments.

Round  2

Reviewer 1 Report

The MS was much improved after the authors revised it.

One suggestion, please consider.

Integrate the answer in the MS, like "The ygl1 is a spontaneous natural mutant, after three generations selfing, the inherited stably ygl1 mutant was used for reciprocally crossing with 1402 to construct the F2 segregating population."

Author Response

One suggestion, please consider.

Integrate the answer in the MS, like "The ygl1 is a spontaneous natural mutant, after three generations selfing, the inherited stably ygl1 mutant was used for reciprocally crossing with 1402 to construct the F2 segregating population."

Response(R)Good suggestion, thanks, it has been revised as suggested. Changes made to the text regarding this are in lines 78 to 80.